# Steps towards a Multiple Myeloma Cure?

**DOI:** 10.3390/jpm12091451

**Published:** 2022-09-03

**Authors:** Alessandro Gozzetti, Monica Bocchia

**Affiliations:** Division of Hematology, University of Siena, Azienda Ospedaliera Universitaria Senese, 53100 Siena, Italy

**Keywords:** multiple myeloma, therapy, minimal residual disease, cure

## Abstract

Multiple myeloma survival has increased in last 20 years because of new treatments, better clinical management due to novel diagnostic tools such as imaging, and better understanding of the disease, biologically and genetically. Novel drugs have been introduced that act with different therapeutic mechanisms, but so have novel therapeutic strategies such as consolidation and maintenance after autologous stem cell transplant. Imaging (such as PET-CT and MRI) has been applied at diagnosis and after therapy for minimal residual disease monitoring. Multiparametric flow and molecular NGS may detect, with high-sensitivity, residual monoclonal plasma cells in the bone marrow. With this novel therapeutic and biological approach, a considerable fraction of multiple myeloma patients can achieve durable remission or even MGUS-like regression, which can ultimately lead to disease disappearance. The big dogma, “Myeloma is an incurable disease”, is hopefully fading.

## 1. Introduction

### 1.1. Multiple Myeloma Revolution: New Drugs

Multiple Myeloma (MM) is still considered an incurable disease in the current literature, although quite a “therapeutic and diagnostic revolution” has happened in last 20 years [1,2,3]. This revolution probably started after the pioneer study that found the drug thalidomide to be effective in relapsed/refractory MM patients (RRMM) [4]. Later, the antiapoptotic and immunomodulatory mechanisms of these drugs (IMIDs) were demonstrated [5,6]. A crucial new concept began to enter hematologists’ minds, i.e., the possibility of novel drugs that act differently from chemotherapy in MM. Drugs that can target not only the monoclonal plasma cell but also the bone marrow microenvironment have been shown to be of importance. Then, the proteasome inhibitor (PI) bortezomib, the second- and third-generation IMIDs lenalidomide and pomalidomide, the second-generation PI carfilzomib, and monoclonal antibodies against CD38, SLAM7, and BCMA were added to the therapeutic armamentarium [7,8,9,10,11,12,13,14,15,16,17,18]. These drugs, combined in triplets and quadruplets, can increase the depth of response during induction therapy; this is subsequently consolidated with autologous stem cell transplant (ASCT), leading to complete responses (CR) in more than 80% of the newly diagnosed transplant-eligible patients (NDTE) [19,20,21,22,23,24,25,26,27]. Progression-free and overall survival (PFS and OS) have rapidly increased from a median of 2–3 years to 6–8 years [2]. In addition, particular forms of aggressive myeloma such as extramedullary or high-risk cytogenetic MM can partially benefit from new drugs [28,29,30,31,32,33,34] and MGRS [35]. Patients with high-risk features, i.e., del 17p, t 4;14, and 14;16, may benefit from new compounds. These patients usually achieve CR after initial treatment, but they usually experience early relapse. Triplet and quadruplet combinations have demonstrated significative benefits when used in HR patients in many clinical trials [19,20,21,22,23,24,25,26,27]. Additionally, elderly patients can benefit from novel drugs that increase disease responses irrespective of age, but still have less eradicating potential than ASCT [36,37,38,39,40,41].

### 1.2. Minimal Residual Disease

The new concept of minimal residual disease (MRD) was introduced in MM by the Spanish group ten years ago with the intent of measuring, with high sensitivity, the disease after therapy [42,43,44,45,46]. It is not enough nowadays, even in clinical practice, to measure responses only using a bone marrow biopsy. Multi-parametric or Next-Generation Flow (NGF) and Next-Generation Sequencing (NGS) are tools recognized by the International Myeloma Working Group (IMWG) to better define CR [47]. Sensitivity has been set by the IMWG at 10^−5^ but many clinical trials are now moving to 10^−6^. MRD has been proven to be the best surrogate for PFS and OS. All new clinical trials put MRD as an endpoint that can quickly measure the depth of response without the need to wait longer for a follow-up to measure PFS and OS. Sustained MRD negativity at 1 year after therapy is an even better prognosticator [48,49,50,51,52,53,54,55,56,57,58,59,60,61,62]. A recent meta-analysis was reported on >8000 MM patients enrolled in 44 studies. MRD negativity, measured using NGF or NGS, was a surrogate for increased PFS and OS [59]. In this study, MRD was always predictive of survival at different sensitivity levels of 10^−4^, 10^−5^, and 10^−6^, but the best cut-off was set at 10^−6^. Outside clinical trials, the standard of care in NDTE patients in Europe has become daratumumab, which is associated with bortezomib thalidomide dexamethasone as an induction regimen to ASCT (Dara-VTD). The study CASSIOPEIA led to its approval. In this study, Dara-VTD was compared to VTD in 1086 patients, and MRD negativity was 64% vs. 44%, respectively [39]. PFS was not reached in the Dara arm vs. 46 months in the VTD arm. Novel triplets and quadruplets are being tested, as is the monoclonal anti-CD38 antibody associated with bortezomib lenalidomide (Dara-VRD, GRIFFIN study, with 51% MRD negativity reported) [60]. Additionally, in other studies MRD-driven therapy is being investigated, with the potential, in the future, to stop or resume therapy based on the absence or the presence of residual disease, respectively [61]. MRD is a predictor of survival irrespective of age. In fact, the best treatment regimen in clinical practice outside trials is now daratumumab plus lenalidomide and dexamethasone (Dara-Rd) from the MAIA trial, which showed MRD negativity in 31% vs. 10% in MM patients with Rd >65 years old who were not eligible for transplant. PFS is not reached at 50 months with Dara-Rd vs. 30 months with Rd [41]. Myeloma is often localized and patchy in the marrow, and limitations may derive from needle aspiration. In the future, peripheral blood could hopefully be considered as a “liquid biopsy” for MRD assessment using new methods such as circulating monoclonal plasma cells, tumor DNA, and mass-spectrometry [62,63,64,65,66,67,68,69,70].

### 1.3. Imaging

Imaging is very important for disease assessment in MM. In fact, it is essential to detect residual disease outside the bone marrow (extramedullary disease). PET-CT and whole-body MRI have been demonstrated as prognosticators for long-term PFS and OS and have become important tools in routine practice after non-Hodgkin lymphomas lessons [71,72,73,74,75,76,77]. MM plasma cells can persist after therapy as focal lesions or as diffuse infiltration in the bone representing an incomplete response to therapy. This residual disease could also be a cause of relapse in MM patients. It is recommended by the IMWG criteria to evaluate imaging after therapy to assess the response [78]. FDG-PET is a functional imaging technique that detects bone disease in MM, and it is optimal for assessing MRD. Imaging combined with MRD detection using NGF or NGS in the bone marrow can provide an even better prognosis. In fact, these techniques combined, when negative, were independent predictors of longer PFS and OS in different randomized clinical trials. Standardization is needed both for PET-CT and MRI and new recommendations have been proposed [78]. However, new tracers and more sensitive imaging methods are needed since pitfalls may exist.

### 1.4. Immunotherapy: Bispecific Antibodies and CART

MM is characterized by a high propensity for infections. This is due to a humorally and cellularly defective immune system [79]. Immune response is also determinant in antitumor activity, and novel therapies are being developed, and some approved, for the treatment of multiple myeloma. Bispecific antibodies (BITEs) that bind myeloma antigens and T-cells have shown significant responses in patients and are multi-refractory to other treatments (PI, IMIDs, and CD38). The anti-BCMA antigen teclistamab, given subcutaneously weekly, led to more than very good partial responses (>VGPR) in 55% of the patients treated with acceptable toxicity [80]. Erlanatamab is another BITE-targeting BCMA and CD3 that led to overall responses in 83% of RRMM patients who were heavily pretreated [81]. Talquetamab is an anti-GPRC5D BITE that has been evaluated and gave VGPR in 53% of the patients treated [82]. CART cell therapy is a new cellular immunotherapy that specifically binds to a myeloma antigen (BCMA) and led to a high overall response rate (64–97%) in heavily pretreated MM patients. Manufacturing time and failures of CART infusion are limitations that need to be ameliorated. Strategies that also consider the timing of CART infusion are being studied [83,84,85].

## 2. Conclusions

The survival progress in MM is evident and patients’ quality of life is being ameliorated. A large number of patients are reaching MRD negativity after therapy. How long this MRD negative status will last is an open question. New therapies and new flow and molecular methods for residual disease detection and imaging will help to better manage patients. Immunotherapies such as BITEs and CART are the new “Kids on the block” and promise to help reach and maintain deep responses. Altogether, these tools will help MM patients to become disease-free and, hopefully, cured.

## Data Availability

Not applicable.

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
