# Peer review of "Steps towards a Multiple Myeloma Cure?"

_jpm, 2022, doi:10.3390/jpm12091451_

Round 1
Reviewer 1 Report
This is a short commentary on a relevant topic in Multiple Myeloma, that is the possibility of achieving a cure in certain group of patients.
I think the manuscript can be improved with few edits.
In the introduction, I will expand the section on patients with high-risk myeloma (lines 38-39) and write MGRS instead of "other gammopathy.
In line 42, I will be more precise than "some years ago".
I will not say "Outside Clinical trials the best standard of care in NDTE patients has become Daratumumab associated to bortezomib thalidomide 56 dexamethasone as induction regimen to ASCT (Dara-VTD)", since there is no direct comparison with other effective regimens such as Dara-VRD. Moreover, Dara-VTD is mainly used in Europe.
In line 78, I will better characterize the importance of MRD and imaging.
I will also check the language, since there are prepositions missing (New Kids on the block, etc) and the style can be improved.
Author Response
Reviewer#1
This is a short commentary on a relevant topic in Multiple Myeloma, that is the possibility of achieving a cure in certain group of patients.
I think the manuscript can be improved with few edits.
- In the introduction, I will expand the section on patients with high-risk myeloma (lines 38-39) and write MGRS instead of "other gammopathy.
we modified and added accordingly, thank you
- In line 42, I will be more precise than "some years ago".
We modified accordingly
- I will not say "Outside Clinical trials the best standard of care in NDTE patients has become Daratumumab associated to bortezomib thalidomide 56 dexamethasone as induction regimen to ASCT (Dara-VTD)", since there is no direct comparison with other effective regimens such as Dara-VRD. Moreover, Dara-VTD is mainly used in Europe.
Sure thank you, it is correct we modified and erased the best and added in Europe
- In line 78, I will better characterize the importance of MRD and imaging.
We added few lines on that paragraph as requested
- I will also check the language, since there are prepositions missing (New Kids on the block, etc) and the style can be improved.
Yes thank you, we went through the manuscript and checked language
Reviewer 2 Report
The authors presented a summary of recent advances in the treatment of multiple myeloma and detection of minimal residual disease. There are a few small notes:
1) On lines 25 and 30, the abbreviation “IMids” is used, while on line 89 – “IMIDs”. Usually the abbreviation “IMiDs” is used.
2) on line 94, “CAR-T” is used, and on lines 96, 97, 103 – “CART”. It's better to bring uniformity.
3) In references [48-62] on line 51, the following articles do not address the use of MRD as a surrogate marker in clinical trials – 50, 56, 58, 59, and 60.
4) In references [66-74] on line 72, the reference 69 “Multiparameter flow cytometric remission is the most relevant prognostic factor for multiple myeloma patients who undergo autologous stem cell transplantation” seems out of place.
5) On line 95, the use of anti-CD19 CAR-T cells is questionable, because in 95% of cases, myeloma cells are CD19-negative.
6) on line 95, the abbreviation ORR is not deciphered.
Author Response
Reviewer 2
The authors presented a summary of recent advances in the treatment of multiple myeloma and detection of minimal residual disease. There are a few small notes:
- On lines 25 and 30, the abbreviation “IMids” is used, while on line 89 – “IMIDs”. Usually the abbreviation “IMiDs” is used.
YES THANK YOU
2) on line 94, “CAR-T” is used, and on lines 96, 97, 103 – “CART”. It's better to bring uniformity.
Yes thank you
- In references [48-62] on line 51, the following articles do not address the use of MRD as a surrogate marker in clinical trials – 50, 56, 58, 59, and 60.
Yes thank you we erased
- In references [66-74] on line 72, the reference 69 “Multiparameter flow cytometric remission is the most relevant prognostic factor for multiple myeloma patients who undergo autologous stem cell transplantation” seems out of place.
We moved reference thank you
5) On line 95, the use of anti-CD19 CAR-T cells is questionable, because in 95% of cases, myeloma cells are CD19-negative.
We deleted CD19 and left BCMA
6) on line 95, the abbreviation ORR is not deciphered. We explained it thanks